# Cisplatin Toxicity Causes Neutrophil-Mediated Inflammation in Zebrafish Larvae

**DOI:** 10.3390/ijms25042363

**Published:** 2024-02-17

**Authors:** Barbara Nunes Padovani, Camila Morales Fénero, Lais Cavalieri Paredes, Mariana Abrantes do Amaral, Omar Domínguez-Amorocho, Marcella Cipelli, Juliana Moreira Mendonça Gomes, Eloisa Martins da Silva, Luísa Menezes Silva, Raquel de Souza Vieira, Mariana Tominaga Pereira, Mario Costa Cruz, Niels Olsen Saraiva Câmara

**Affiliations:** 1Department of Immunology, Institute of Biomedical Sciences, University of Sao Paulo, São Paulo 05508000, Brazil; barbara.padovani@usp.br (B.N.P.); lais.cavalieriparedes@icb.usp.br (L.C.P.); oadominguez80@gmail.com (O.D.-A.); marcella.cipelli@usp.br (M.C.); luisamenezes@usp.br (L.M.S.); raquel_vieira@usp.br (R.d.S.V.); mariana.tominaga@usp.br (M.T.P.); mccruz@usp.br (M.C.C.); 2Department of Microbiology and Environmental Toxicology, Biomedical Sciences, University of California Santa, Santa Cruz, CA 95064, USA; 3Nephrology Division, Department of Medicine, Federal University of Sao Paulo, São Paulo 04023062, Brazil; marianaamaral1231@gmail.com (M.A.d.A.); eloisasilva456@gmail.com (E.M.d.S.); 4Department of Developmental & Molecular Biology, Albert Einstein College of Medicine, New York, NY 10461, USA; julianammgomes@gmail.com

**Keywords:** neutrophil, *Danio rerio*, AKI

## Abstract

Cisplatin is an antineoplastic agent used to treat various tumors. In mammals, it can cause nephrotoxicity, tissue damage, and inflammation. The release of inflammatory mediators leads to the recruitment and infiltration of immune cells, particularly neutrophils, at the site of inflammation. Cisplatin is often used as an inducer of acute kidney injury (AKI) in experimental models, including zebrafish (*Danio rerio*), due to its accumulation in kidney cells. Current protocols in larval zebrafish focus on studying its effect as an AKI inducer but ignore other systematic outcomes. In this study, cisplatin was added directly to the embryonic medium to assess its toxicity and impact on systemic inflammation using locomotor activity analysis, qPCR, microscopy, and flow cytometry. Our data showed that larvae exposed to cisplatin at 7 days post-fertilization (dpf) displayed dose-dependent mortality and morphological changes, leading to a decrease in locomotion speed at 9 dpf. The expression of pro-inflammatory cytokines such as interleukin *(il)-12*, *il6*, and *il8* increased after 48 h of cisplatin exposure. Furthermore, while a decrease in the number of neutrophils was observed in the glomerular region of the pronephros, there was an increase in neutrophils throughout the entire animal after 48 h of cisplatin exposure. We demonstrate that cisplatin can have systemic effects in zebrafish larvae, including morphological and locomotory defects, increased inflammatory cytokines, and migration of neutrophils from the hematopoietic niche to other parts of the body. Therefore, this protocol can be used to induce systemic inflammation in zebrafish larvae for studying new therapies or mechanisms of action involving neutrophils.

## 1. Introduction

Cisplatin, also known as *cis*-diamminedichloroplatinum (II), is a chemotherapy drug used to treat various types of cancer, including head, neck, sarcomas, testicles, and ovaries [1,2,3]. Its antitumor effect is due to its cytotoxic properties, which involve interacting with deoxyribonucleic acid (DNA) and activating cell death pathways [4,5]. The drug’s toxicity is associated with oxidative stress and reactive oxygen species (ROS) release, affecting cellular components such as mitochondria [3]. The elevated levels of cell death contribute to the development and progression of inflammation by releasing pro-inflammatory mediators, including interleukin (IL)-1β, tumor necrosis factor (TNF)-α, IL-6, and IL-8 [6,7,8,9]. These cytokines are released by various cell types, including resident immune cells and endothelial cells, and result in leukocyte recruitment [6]. Neutrophils are often the first responders, rapidly reaching the damaged site and playing a role in phagocytosis, the release of neutrophil extracellular traps (NETs), and the production of inflammatory mediators and granules containing reactive species [10,11]. However, due to its lack of specificity, cisplatin can cause various side effects in patients, including gastrointestinal disorders, hypersensitivity reactions, and severe kidney damage [3,12,13]. The effects can also extend to significant neurotoxicity and ototoxicity, which are commonly observed in individuals undergoing cisplatin therapy [14]. Neurotoxicity results in damage to the nervous system and can lead to sensory and motor impairments, while ototoxicity affects the auditory system and compromises hearing functionality [15,16]. Cisplatin can enter cells through passive diffusion or membrane transporters like copper uptake protein 1 (Ctr1) and organic cation transporter 2 (OCT2) [15,17,18]. Once inside the cell, the drug interacts with DNA, activating various signal transduction pathways and disrupting the cell cycle in cancer cells [19]. The mechanisms of action of cisplatin are still being studied, but it is known to activate intrinsic and extrinsic pathways of apoptosis and induce oxidative stress, leading to cytotoxicity [20,21]. Accumulation of cisplatin in the kidneys contributes to cell death and triggers inflammation, particularly in the proximal tubular segment of the nephron, where cisplatin accumulates the most [22]. This process can result in kidney function loss and the development of acute kidney injury (AKI), characterized by proximal tubule damage and inflammatory response [23]. The inflammation caused by cisplatin can be enhanced by the generation of damage-associated molecular patterns (DAMPs) that are recognized by pattern recognition receptors (PRRs), activating different signaling pathways and recruiting transcription factors like nuclear factor-κB (NF-κB) and interferon regulatory factor (IRF)-3, which in turn induce the production of pro-inflammatory cytokines, including IL-8 and IL-6 [7,24].

Due to its nephrotoxic effects, cisplatin is commonly used as a model for studying acute kidney injury (AKI) in animal models [25]. In this regard, the zebrafish (*Danio rerio*) has emerged as a valuable model for studying human diseases due to its rapid development, high genetic similarity to humans (70%), high external fertilization rates, conservation of physiological and inflammatory processes, and low maintenance cost [26,27]. The zebrafish’s ability to regenerate its kidney has made it particularly useful for studying kidney injury in the research literature [28,29]. Several protocols have been developed to induce AKI models in zebrafish larvae closely resembling human AKI characteristics. For example, microinjection of gentamicin into larvae can lead to edema formation, loss of cell polarity, and disruption of the proximal tubule [30]. Microinjection of cisplatin into 48 h post-fertilization (hpf) larvae causes cellular vacuolization, tubular lumen distension, flattening, and loss of the brush border, ultimately compromising renal function in these fish [31]. The challenge of studying the effect of cisplatin on zebrafish larvae is that these animals are still in their developmental stages, including the development of their adaptive immune system [32]. As a result, studies with larvae primarily focus on the innate immune system.

Models of cisplatin-induced AKI in adult zebrafish have demonstrated loss of tubular structure, increased inflammatory infiltrate, and a high rate of cell death in the kidney [33]. The injury caused by cisplatin appears to be evolutionarily conserved between teleosts and humans, making zebrafish an intriguing organism for studying the renal inflammation induced by this drug [31]. However, most studies primarily focus on the kidney effects of cisplatin and overlook any potential systemic effects. This study aims to examine the inflammatory impact of cisplatin exposure on whole larvae to understand better the diverse outcomes associated with using this drug in the entire body.

## 2. Results

### 2.1. Cisplatin Induces Dose-Dependent Mortality

Previous studies in cells, mice, and zebrafish have demonstrated a dose-dependent effect of cisplatin [33,34,35]. To investigate whether administering cisplatin in the embryonic medium would impact larval survival, we exposed 7-day post-fertilization (dpf) wild-type larvae to increasing concentrations of cisplatin (0.025 mg/mL, 0.05 mg/mL, 0.10 mg/mL, 0.15 mg/mL, and 0.25 mg/mL). Cisplatin was directly diluted in the embryonic medium, and larval survival was monitored up to 10 dpf. Our results revealed no significant mortality within 24 h of exposure to any cisplatin concentration (Figure 1). After 48 h, the lowest concentration (0.025 mg/mL) did not show significant mortality. However, mortality was significant and dose-dependent at concentrations of 0.05 mg/mL (26.7%), 0.10 mg/mL (45%), 0.15 mg/mL (71.7%), and 0.25 mg/mL (99.4%). By 72 h, these concentrations induced 100% mortality.

### 2.2. Cisplatin Reduces Movements and Induces Morphological Deformities in Larvae

After exposure to 0.10 mg/mL of cisplatin (LD50—lethal dose, 50%), alterations in the curvature of the rostral–caudal body axis of the larvae were noted after 48 h. These alterations were not observed in age-equivalent control animals (Figure 2A). Quantification of larval movements showed that the use of cisplatin resulted in a significant decrease in the speed of larval locomotion at 9 dpf compared to the control group (Figure 2B). We hypothesize that the reduced locomotor activity is related to the morphological changes caused by cisplatin.

### 2.3. Systemic Expression of Pro-Inflammatory Cytokines Is Increased after Exposure to Cisplatin

Because we observed a decrease in survival in larvae exposed to cisplatin, we decided to evaluate the inflammatory profile of these animals after 48 h of exposure. As such, we assessed the expression of cytokines in the whole larvae using quantitative polymerase chain reaction (qPCR). Our data showed a significant increase in the expression of the pro-inflammatory cytokines *il6*, *il8*, and *il12* in animals exposed to cisplatin compared to the control group (Figure 3). Specifically, the expression of *il8* and *il6* in the cisplatin-treated group was two times higher, highlighting the inducing role of cisplatin in inflammation. We also analyzed the expression of the anti-inflammatory cytokine *il10* and the kidney injury molecule 1 (*kim1*); however, no significant difference was observed between the groups.

### 2.4. Neutrophil Frequency Is Increased after Exposure to Cisplatin

Based on the increase in expression of *il8* (Figure 3), which has been described as a chemokine for neutrophils [36], we analyzed the total number of neutrophils in the whole larvae using flow cytometry. Figure 4A shows the gate strategy used in the control and treated groups. After 2 days of cisplatin exposure, we did not observe a significant difference in the percentage of positive lysozyme cells between the two groups. However, there was a significant increase in the frequency of CD11b+/Ly6G+ neutrophils in animals treated with cisplatin (Figure 4B). This correlates with our previous result, where we observed an increase in *il8* expression.

### 2.5. Cisplatin Generates a Decrease in Neutrophils in Pronephros

Next, because we observed an increase in Cd11b/Ly6G neutrophils and an increase in the neutrophil chemoattractant *il8* in the whole larvae, we wondered whether neutrophils were moving to specific regions. To evaluate the distribution of neutrophils, we used the transgenic strain Tg(lysC:DsRed2)^nz50^ with a red fluorescent tag in neutrophils and quantified the accumulation of these cells as an indication of inflammation. Analyzing the distribution of fluorescent neutrophils, we noted a decrease in cells in the region of the larvae’s pronephros (Figure 5A, white circle). Because cisplatin is known to accumulate in the kidney, we thought that the accumulation of neutrophils could be due to the damage caused by cisplatin [23]. Unexpectedly, larvae exposed to cisplatin showed a significant decrease in MFI in the glomerular region of the pronephros (Figure 5B).

### 2.6. Cisplatin Exposure Induces Cell Death in the Pronephros

The accumulation of cisplatin in the kidneys generates high levels of apoptosis in the organ, which is an important feature for the development of AKI. Thus, we analyzed whether this pattern was occurring in the glomerular region of the pronephros using the terminal deoxynucleotidyl transferase (TdT) dUTP nick end labeling (TUNEL) assay. As shown in Figure 6, larvae exposed to cisplatin displayed more apoptotic cells than the controls in the quantified region (circled in white). However, apoptotic cells can be observed in other body areas in Figure 6A. The results of this experiment suggest that the decrease in the MFI of neutrophils in the glomerular region of the pronephros, as shown in Figure 5, is due to cell death caused by cisplatin.

### 2.7. Cisplatin Induces Systemic Inflammation 

Our previous data showed a decrease in the neutrophil population in the pronephros (Figure 5) and an increase in the number of apoptotic cells in this region (Figure 6). Considering these findings, we evaluated the infiltration of neutrophils in different tissues and organs after the administration of cisplatin compared to the control group. Figure 7A displays representative images of both groups, showing a higher presence of neutrophils in larvae treated with cisplatin. Our data reveal that approximately 85% of control larvae show no signs of inflammation (Figure 7B). Among the larvae exposed to 0.10 mg/mL of cisplatin, 60% were classified as having mild inflammation, while 25% exhibited severe inflammation. These results suggest that the administration of cisplatin in the embryonic medium leads to systemic inflammation involving a significant influx of neutrophils.

## 3. Discussion

Cisplatin is used as an inducer of acute kidney injury in animal models. This drug is directly toxic to kidney cells and induces activation of inflammatory pathways [37]. The response to renal injury is evolutionarily conserved between humans and zebrafish. As observed in our results, the method of administering the drug may alter the commonly observed effect in AKI models using this drug [38]. The existing protocols for generating AKI are based on injecting the drug and controlling the solution intraperitoneally or intravenously. We chose to dilute cisplatin in the embryonic medium and evaluate the generation of kidney damage and systemic inflammation. 

Studies have shown that cisplatin exhibits a cumulative effect consistent with the pattern observed in our data [23]. As the duration of exposure to the drug increased, we observed an associated increase in mortality. The animals showed a dose-dependent response, as reported in other studies (Figure 1) [33,39,40]. Another observed effect was decreased movement of the larvae exposed to cisplatin (Figure 2). In addition to being toxic to the kidneys, this drug also presents ototoxicity for users of this antineoplastic agent [15]. In zebrafish, this ototoxicity has been associated with the release of ROS and activation of cell death pathways, resulting in changes in the larvae’s lateral line and ear hair cells [41,42]. These structures are responsible for responding to mechanical disturbances caused by movement in water and transmitting this information to the brain to generate a behavioral response associated with detecting prey or escaping from predators [43]. Cisplatin can affect the hair cells of the zebrafish lateral line and consequently alter locomotion behavior [12]. Studies evaluating drug toxicity have shown that degeneration of the notochord may occur, resulting in longitudinal shortening of the body and decreased locomotor response to stimuli, similar to our results [44,45]. The development of the larva may be compromised since the entire body surface of the larva is in contact with cisplatin.

This drug produces and releases inflammatory mediators by resident immune cells and renal epithelial cells [11]. In general, cells that are damaged or in the process of apoptosis release DAMPs, such as genomic DNA, mitochondrial DNA, and adenosine triphosphate (ATP). After recognizing these DAMPs, an inflammatory cascade is activated, resulting in the recruitment of leukocytes to the damaged site [46]. As seen in Figure 3, there was an increase in the pro-inflammatory cytokines *il6*, *il12*, and *il8*. Together, these cytokines assist in activating and recruiting cells of myeloid and lymphoid origin, such as neutrophils, macrophages, natural killer cells, and T cells [47,48,49]. IL-6 plays a role in regulating neutrophils during inflammation [50]. A study using a nephrotoxin-induced AKI model showed that kidney injury induces local and systemic expression of *il6*, and this increase is associated with neutrophilic infiltration and exacerbation of the inflammatory response [51]. The IL-8 chemokine is extremely important for neutrophil chemotaxis. Our data show that the significant increase in *il8* in the group exposed to cisplatin is accompanied by increased proliferation and activation of this cell type (Figure 4). Our data also showed that cisplatin affected the numbers of neutrophils in the pronephros (Figure 5) and increased the numbers of apoptotic cells in the same region (Figure 6). 

Neutrophils, as cells of innate immunity, play a significant role in the inflammation generated by cisplatin. These phagocytes release granules containing reactive species and NETs, contributing to systemic inflammation [52]. A study using another AKI inducer, gentamicin, showed that zebrafish larvae had increased neutrophils at 2 days post-injection (dpi) [53]. These data corroborate our findings, showing that drug toxicity can lead to neutrophil-mediated inflammation in zebrafish. However, the dilution of the drug directly in the embryonic medium resulted in systemic inflammation in the larvae (Figure 7), which was not restricted to the pronephros as observed in models of acute kidney injury in adult zebrafish. In addition, in Figure 3, we observed no significant increase in *kim1*, which was expected to increase after the stimulus. However, studies have reported that urinary levels of KIM-1, a kidney damage marker, in response to cisplatin treatment depend on the duration of exposure. Therefore, significant increases in KIM-1 expression may only occur after exposure to cisplatin for more than 48 h [54].

Our research acknowledges limitations arising from using a single concentration of cisplatin and a specific time point of exposure. The absence of comparisons with alternative models or drugs limits the generalizability of our findings. Additionally, not assessing cisplatin or its metabolite levels in larvae restricts our understanding of dose–response relationships and toxicity mechanisms. Future studies will focus on analyzing doses, duration of exposure, and using other models. Further investigation into molecular pathways mediating cisplatin’s effects on cytokines, neutrophils, cell death, and kidney damage is crucial for identifying potential intervention targets. 

In conclusion, cisplatin induces systemic inflammation in zebrafish larvae, as evidenced by increased cytokine expression and neutrophil migration. Morphological changes and cell death suggest a broader toxicity profile (Figure 8). Although our data indicate that the drug affects the renal system, using cisplatin dilution in the embryonic medium as a model of systemic inflammation could be more effective and straightforward, particularly for studying innate immunity, especially in neutrophils. Models of systemic inflammation are interesting tools for identifying mechanisms of action and potential new therapies that can assist in managing the inflammatory process.

## 4. Materials and Methods

### 4.1. Animals

Embryos of the AB and Tg(lysC:DsRed2)^nz50^ strains were provided and were maintained at the Zebrafish Animal Facility of the Institute of Biosciences of the University of São Paulo (USP) [55]. The embryos were obtained through natural spawning, collected in Petri dishes with embryo medium E3 (5 mM NaCl, 0.17 mM KCl, 0.33 mM CaCl2, 0.33 mM MgSO4, pH 7.2) and methylene blue, and kept in the incubator (Sanyo, Moriguchi, Osaka, Japan) at 28.5 °C with a light/dark cycle of 14/10 h. All protocols used in this study were approved by the ethics committee in animal research of the Institute of Biomedical Sciences and the Institute of Biosciences of the University of São Paulo (8619291019).

### 4.2. Analysis of the Survival of Zebrafish Larvae in Response to Cisplatin

Cisplatin (*cis*-Diammineplatinum (II) dichloride) was purchased from Sigma-Aldrich, St. Louis, MO, USA, and was prepared following the manufacturer’s instructions. Cisplatin working solutions were prepared in E3 medium at final concentrations of 0.025, 0.05, 0.10, 0.15, and 0.25 mg/mL for the experiments. Cisplatin was added to the embryonic medium of the larvae at 7 days post-fertilization (dpf) and was maintained until the end of the experiment. The larvae were placed in 6-well plates (Costar 3516, Corning Incorporated, New York, NY, USA) with 20 larvae per well. The pH of the medium was measured to verify that there was no variation after adding the drug, and it remained between 7.2 and 7.3. The larvae survival was monitored daily from 7 dpf until 10 dpf. Following the LD50 determination, further analyses were conducted using a cisplatin concentration of 0.10 mg/mL with an exposure duration of 48 h.

### 4.3. Analysis of Locomotor Activity 

Larvae were examined 48 h after exposure to cisplatin. The recording was carried out on a 96-well plate (Costar 3599, Corning Incorporated, New York, NY, USA) with one larva added per well. The larvae were maintained in 100 µL of embryonic medium. Data acquisition was performed using the ZEISS Axio Zoom.V16, with settings of two frames per second, a magnification of 7x, and a time interval of 10 min. The videos were analyzed using ImageJ 1.54g software with the TrackMate plugin, resulting in the average velocity of each larva (microns/second).

### 4.4. Gene Expression Analysis

Pools of 20 AB larvae from the control and cisplatin (0.10 mg/mL) groups were collected and euthanized with 0.3 mg/mL of Tricaine (MS-222) (Sigma-Aldrich, St. Louis, MO, USA) [56,57]. RNA extraction was performed using TRIzol (Invitrogen, Carlsbad, CA, USA), following the manufacturer’s instructions. RNA concentration was measured by absorbance in NanoDrop (Thermo Fisher, Waltham, MA, USA). Then, cDNA was synthesized from 2 μg of RNA and diluted (1:10). In the PCR reaction, 4 μL of cDNA, 0.5 μL of each primer (125–500 nM), and Power Master Mix Syber (Thermo Fisher, Waltham, MA, USA) were used. Relative gene expression was calculated using elongation factor 1-alpha (*efl1a1*) as a reference gene and analyzed through the 2(−ΔΔCT) method [58]. The Sybr primers (Exxtend biotecnologia Ltd., Paulínia, Brazil) used are shown in Table 1.

### 4.5. FACS Analysis

To quantify the number of neutrophils, samples were collected in pools containing 50 Tg(lysC:DsRed2)^nz50^ transgenic larvae, where lysozyme is marked with red fluorescent protein labeling. The larvae were euthanized as previously described and maintained in a 2% fetal bovine serum (FBS)/1X PBS (phosphate-buffered saline) (Thermo Fisher, Waltham, MA, USA) solution at 4 °C until processing. Subsequently, the samples were macerated in a 40 µm cell strainer (Corning Incorporated, New York, NY, USA) and resuspended in 2% FBS/1X PBS. Cells were incubated with CD11b-PE/Cyanine7 (BioLegend, San Diego, CA, USA) and Ly6G-APC (BD Biosciences, San Jose, CA, USA) antibodies. The samples were examined on a BD FACSCanto II and analyzed using FlowJo v10 (Tree Star) software.

### 4.6. Quantification of Mean Fluorescence Intensity

The quantification of mean fluorescence intensity (MFI) indicated kidney inflammation. Tg(lysC:DsRed2)^nz50^ larvae were anesthetized with 0.168 mg/mL of Tricaine (MS222) (Sigma-Aldrich, St. Louis, MO, USA) and placed on a microscope slide with 3% methylcellulose [56,57]. The animals from the control and treated groups were carefully manipulated for lateral vision, and images were captured using the ZEISS Axio Zoom.V16. The MFI of neutrophils in the glomerular region of the pronephros was quantified using ImageJ software.

### 4.7. Cell Death Assay

Samples were fixed with 4% paraformaldehyde (PFA) overnight at 4 °C, washed with 1X PBS, and dehydrated in a series of methanol/1X PBS washes: 25%, 50%, 75%, and 100%, each for 5 min. The samples were kept in 100% methanol at −20 °C until use. For determining cell death, the samples were rehydrated with decreasing concentrations of methanol/1X PBS (75%, 50%, and 25%), followed by three PBT washes (0.5% Triton/1X PBS) for 5 min each. Bleaching with 2% H2O2 and 0.5% KOH was performed for 30 min, followed by three PBT washes. Permeabilization was achieved using 1 mg/mL of type IV collagenase (Thermo Fisher, Waltham, MA, USA) for 60 min. The In Situ Cell Death Detection TMR RED enzyme kit (Roche, Basel, Switzerland) was used following the manufacturer’s instructions. The kit is based on detecting breaks in single- and double-stranded DNA that occur in the early stages of the apoptosis process. Cell nuclei were stained with Hoechst (Thermo Fisher, Waltham, MA, USA) (1:1000). The larvae were mounted in a lateral position on a 1% low melting point agarose. The images were captured with an LSM 780 NLO confocal microscope and analyzed using ImageJ software’s “Cell Counter” plugin.

### 4.8. Systemic Inflammation Score

Tg(lysC:DsRed2)^nz50^ larvae were anesthetized, placed on a microscope slide with 3% methylcellulose (Sigma-Aldrich, St. Louis, MO USA), and positioned laterally to visualize the entire larvae. Images were captured using the ZEISS Axio Zoom.V16 microscope. The larvae were classified as none, mild, or severe for systemic inflammation based on the criteria described in Table 2 [59].

### 4.9. Statistics

The data were presented as mean and standard deviation. The survival curve was evaluated using the log-rank test (Mantel–Cox) and the Gehan–Breslow–Wilcoxon test. The difference between the two experimental groups was determined using a *t*-Student test. The chi-square test was used to analyze systemic inflammation. A *p*-value of less than 0.05 was considered significant for all graphs. All graphs were created using GraphPad Prism^®^ 6.

## Figures and Tables

**Figure 1 ijms-25-02363-f001:**
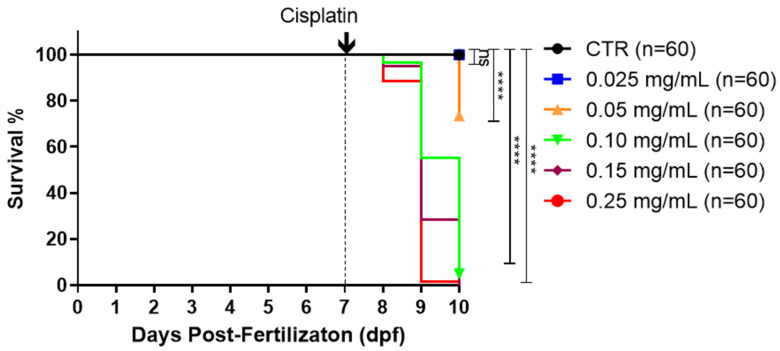
Cisplatin induces dose-dependent mortality. At 7 dpf, the larvae were exposed to increasing concentrations of cisplatin (0.025 mg/mL, 0.05 mg/mL, 0.10 mg/mL, 0.15 mg/mL, and 0.25 mg/mL) or kept in the embryonic medium (CTR). The survival of the larvae was monitored daily from 7 to 10 dpf. Statistical differences were assessed using the log-rank test (Mantel–Cox) (**** *p* ≤ 0.0001, ns = not significant (*p* > 0.05)). Data from two independent experiments, each containing 60 larvae per group.

**Figure 2 ijms-25-02363-f002:**
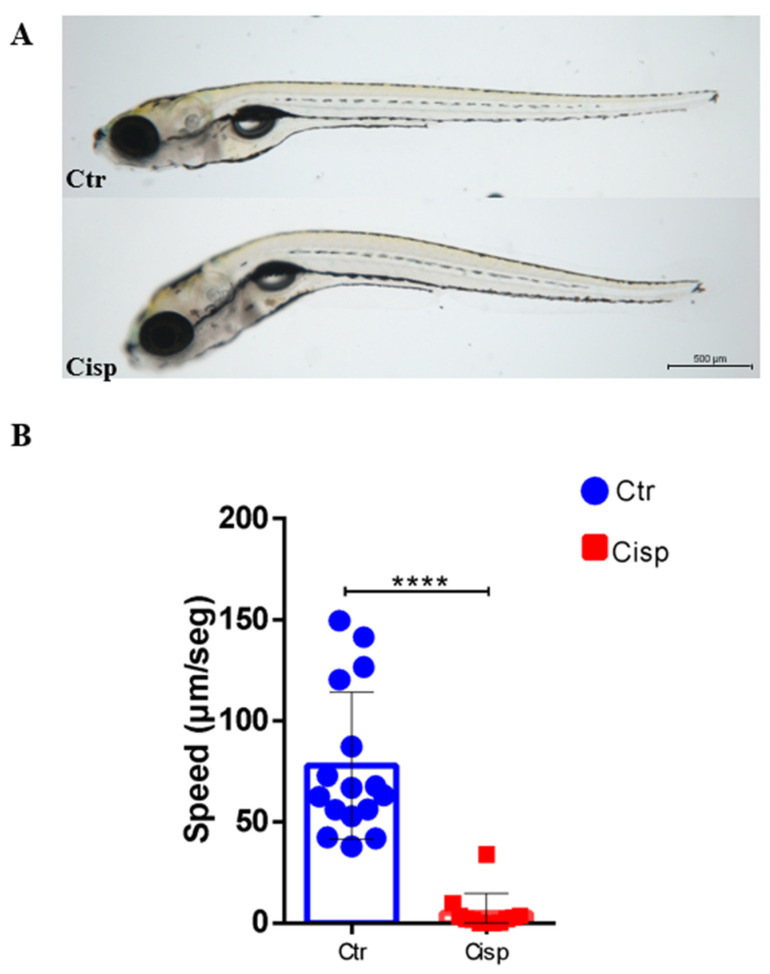
Cisplatin affects the larvae’s curvature and decreases the locomotion speed. Cisplatin (Cisp) was added to the embryonic medium containing the larvae at 7 dpf and maintained for 48 h. Control larvae (Ctr) were placed only in the embryonic medium. (**A**) Cisplatin-induced alterations in the larvae curvature after 48 h of exposure compared to the control group. Rostral on the left, dorsal above. Magnitude 30×. (**B**). Cisplatin induced a significant decrease in larval movements. The locomotor activity of the larvae was recorded for 10 min, and the velocity (μm/s) was quantified. Statistical differences were evaluated using the *t*-Student test (**** *p* ≤ 0.0001).

**Figure 3 ijms-25-02363-f003:**
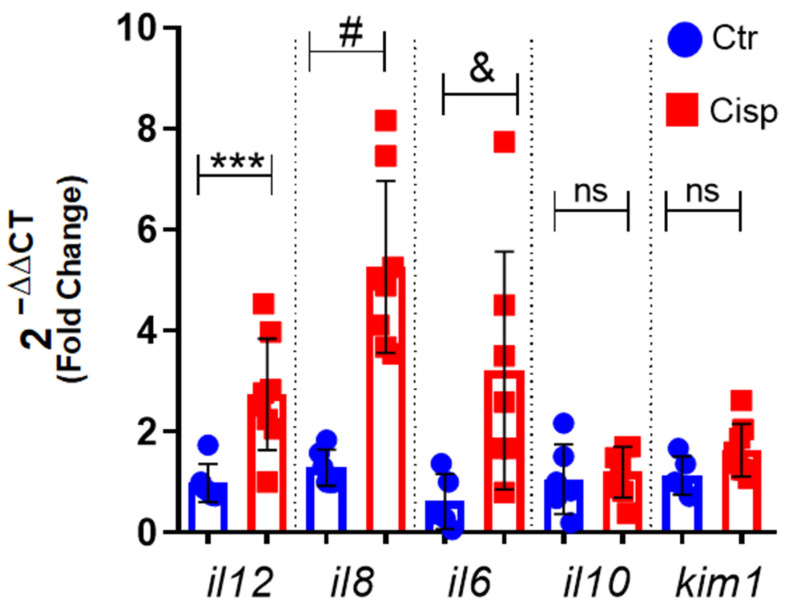
Cisplatin increases gene expression of pro-inflammatory cytokines. The relative gene expression of pro-inflammatory cytokines (*il12, il8,* and *il6*), anti-inflammatory cytokine (*il10*), and renal damage marker (*kim1*) was measured in larvae at 9 dpf. The larvae were exposed to 0.10 mg/mL of cisplatin (Cisp) or embryonic medium (Ctr) for 48 h. The data show that cisplatin results in increased expression of inflammatory markers. Means were obtained from 6 to 9 samples, which referred to a pool of 20 larvae per sample. A *t*-Student test was performed between the cisplatin groups and the control group for each gene, with significance indicated as *** *p* = 0.0030; # *p* = 0.0001; and & *p* = 0.0382; ns = not significant, *p* > 0.05. The data are presented as mean and standard deviation.

**Figure 4 ijms-25-02363-f004:**
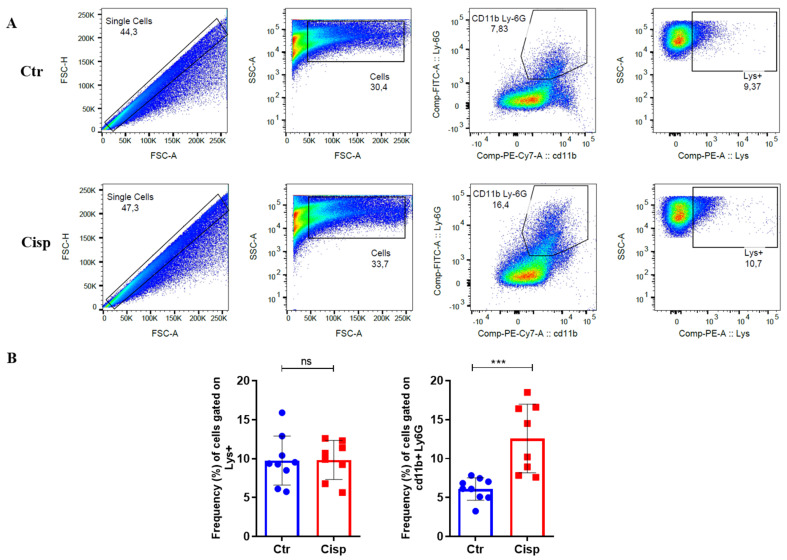
Cisplatin increases the frequency of neutrophils all over the body. The percentage of lysozyme+ cells and neutrophils was measured in larvae after 48 h of cisplatin exposure or embryonic medium (Ctr). (**A**). Representative figures show the gating strategy used for flow cytometry analysis, demonstrating the selection of different cell populations (neutrophils) using CD11b and Ly6G antibodies or lysozyme fluorescence. FACs (fluorescence-activated cell sorting) analysis was conducted using a pool of 50 Tg larvae Tg(lysC:DsRed2)^nz50^. (**B**). The frequency of neutrophils is increased in the cisplatin-treated group compared to the control larvae. The mean was obtained from 8 to 10 samples, and the *t*-Student test was performed between the groups (*** *p* = 0.0008, ns = not significant (*p* > 0.05)). Data from the two experiments are presented as mean and standard deviation.

**Figure 5 ijms-25-02363-f005:**
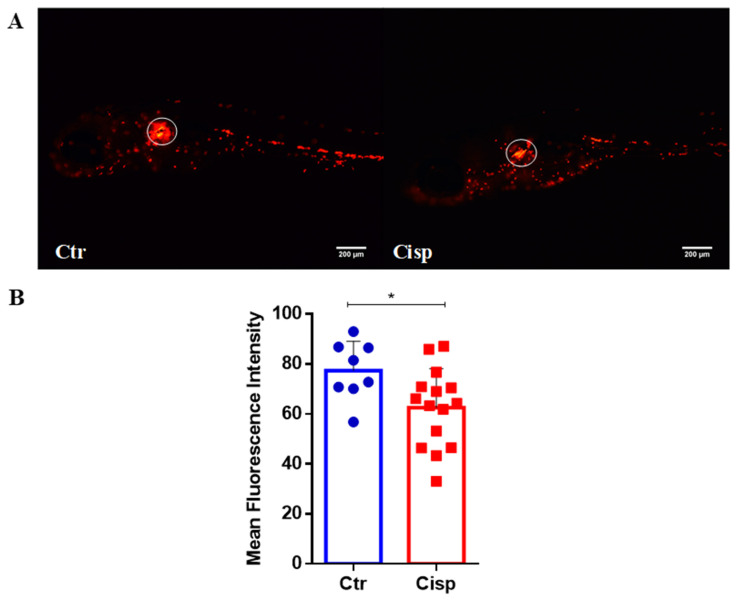
Cisplatin decreases neutrophils in the glomerular pronephros region. (**A**). Representative images of Tg(lysC:DsRed2)^nz50^ larvae at 9 dpf, controls (Ctr), and those exposed to 0.10 mg/mL of cisplatin (Cisp) for 48 h. Rostral on the left, dorsal above. (**B**). The quantification of the mean fluorescence intensity (MFI) of the glomerular region of the pronephros (circled in the blank in (**A**)) shows a decrease in MFI in the cisplatin-treated group compared to the control group. Statistical differences were evaluated using the *t*-Student test (* *p* = 0.0292). Data are presented as mean and standard deviation.

**Figure 6 ijms-25-02363-f006:**
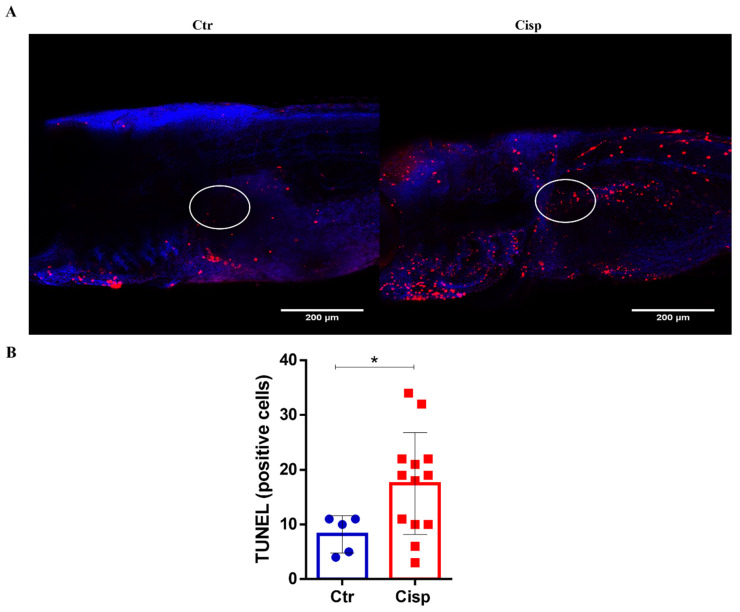
Cisplatin exposure induces cell death in pronephros. Larvae AB (wild type) at 9 dpf were exposed to 0.10 mg/mL of cisplatin (Cisp) or embryonic medium (Ctr) for 48 h. The number of positive cells in the glomerular region of the pronephros indicating cell death was quantified using the TUNEL assay. (**A**). Representative images showing cell nucleus marking with Hoechst (in blue) and cell death marking (in red). Rostral on the left, dorsal above. (**B**). The number of TUNEL-positive cells in the glomerular region of the pronephros (circled in white in (**A**)) was increased. The *t*-Student test was performed between the groups (* *p* = 0.0483). Data are presented as mean and standard deviation.

**Figure 7 ijms-25-02363-f007:**
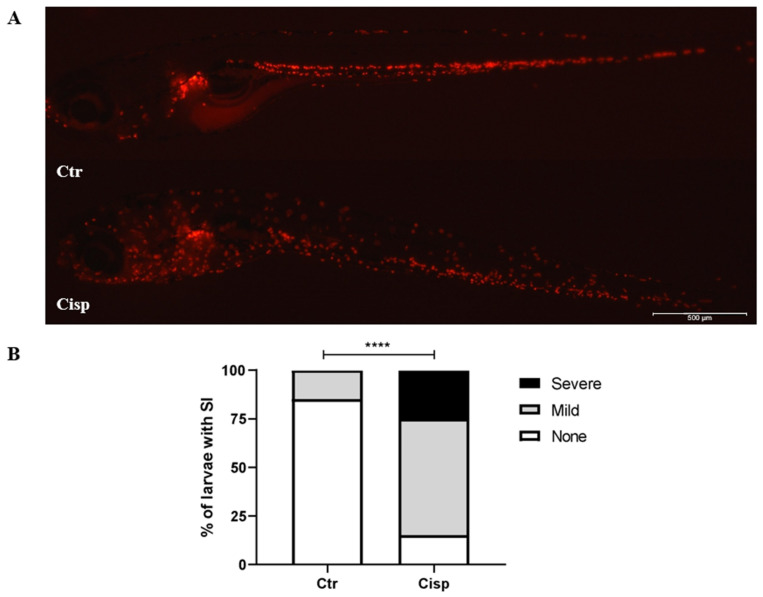
Cisplatin induces neutrophil infiltration to different tissues and organs causing mild/severe inflammation. (**A**). Representative images of Tg(lysC:DsRed2)^nz50^ larvae at 9 dpf exposed for 48 h to 0.10 mg/mL of cisplatin (Cisp) or embryonic medium (Ctr). Rostral on the left, dorsal above. (**B**). Percentage of larvae with systemic inflammation (SI) in the whole larvae in Ctr (N = 20) and Cisp (*n* = 20). Based on the SI score, each larva was classified into three groups: none, mild, and severe. Statistical differences were evaluated using the chi-square test (**** *p* ≤ 0.0001) to analyze SI scores.

**Figure 8 ijms-25-02363-f008:**
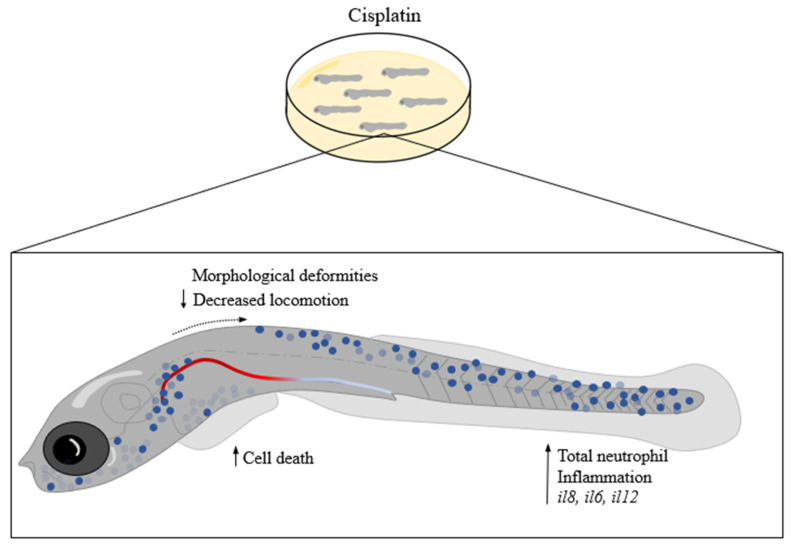
Cisplatin causes systemic inflammation mediated by neutrophils. Larvae at 9 dpf exposed for 48 h at 0.10 mg/mL of cisplatin show morphological alterations and decreased locomotion speed. Cisplatin increased inflammatory cytokines (*il8*, *il6*, and *il12*), number of neutrophils, and increased cell death.

**Table 1 ijms-25-02363-t001:** List of primers used for qPCR. The genes analyzed are described in the first column. The primers forward and reverse are next, respectively.

Gene	Forward (5′ -> 3′)	Reverse (5′ -> 3′)
*efl1a1*	AGTGTTGCCTTCGTCCCAAT	TTCCATCCCTTGAACCAGCC
*il6*	GGCATTTGAAGGGGTCAGGA	TCAGGACGCTGTAGATTCGC
*il8*	GTTTTCCTGGCATTTCTGACCA	GCGTCGGCTTTCTGTTTCAA
*il12*	ACGCAAACGGTGTCTGTCT	CTCTGTAGGCATTCGCTCTCAT
*havcr1/kim1*	CGCTAGAAGTAAGGCAGAA	CACTGTTCGTATTCGCTTTC

**Table 2 ijms-25-02363-t002:** Criteria employed for the classification of systemic inflammation. The scoring system (none, mild, and severe) was based on the distribution of neutrophils in zebrafish.

Score	Distribution of Neutrophils
None	Neutrophils aligned in the caudal hematopoietic tissue (CHT); neutrophils concentrated in the glomerular region of pronephros; absence of cells in the heart, brain, and around the eyes.
Mild	More dispersed neutrophils are in the glomerular region of pronephros; neutrophils are present in the muscular portion of the tail; cell infiltration in the liver.
Severe	Low concentration of cells in the glomerular region of pronephros; a high number of neutrophils in the liver; presence of neutrophils in the heart, brain, and around the eyes; CHT emptied.

## Data Availability

The data presented in this study are available on request from the corresponding author.

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
