# Peer review of "Cisplatin Toxicity Causes Neutrophil-Mediated Inflammation in Zebrafish Larvae"

_ijms, 2024, doi:10.3390/ijms25042363_

Round 1

Reviewer 1 Report

Comments and Suggestions for Authors

Cisplatin toxicity causes neutrophil-mediated inflammation in zebrafish larvae

Summary

The above-mentioned article deals with the systemic effect of the chemotherapeutic agent cisplatin on the organism of a zebrafish larva with the aim of extending previous models, which refer exclusively to AKI, to investigate the effect of the inflammatory reaction triggered by cisplatin not only on the kidney, but also in the entire organism. The investigations revealed a dose-dependent mortality of the larvae, as well as morphological changes and decreased locomotion.  In addition, pro-inflammatory cytokines and neutrophils and their migration in connection with cisplatin exposure were examined in more detail.

The article was subdivided according to the guidelines. The illustrations are mostly clearly arranged and visualize the results.

The author uses current and a variety of sources from different authors.

All abbreviations used should be explained in the text. Otherwise, a list of abbreviations should be created. Common abbreviations must also be explained.

For example:

    Line 96: LD50

    Line 343: PBS

Introduction

The introduction gives a good insight into the necessary subject matter. However, the side effect profile of cisplatin should be introduced in more detail. If the common side effects are mentioned, the most frequent ones should also be named (in addition to nephrotoxicity, neurotoxicity and ototoxicity), which in the end are the results of the processes mentioned, such as inflammation. Especially since ototoxicity is also a part of the discussion, it would be advisable to include it in the introduction.

Results

In the results section, few to almost no results are presented as numeric values, which makes the statements made less clear and comprehensible. The values for the diagrams shown cannot be found in the text. This should be supplemented with all p-values and standard deviations. All p-values, even if they are not significant, should also be presented. Most of the sections in the results section are very short and consist only of an introduction to the topic and an already completed interpretation and formulation of hypotheses. This should be part of the discussion and not appear in this part of the article. The relationship between the actual result and the interpretation or introduction is not correct.

For example:

    2.3: Lines 128 to 132 correspond to an introduction to the topic, followed by two sentences on the results. No absolute or relative numbers are used, although the corresponding Figure 3 is extensive and shows that many numerical values must be present.

Figure 1: The relevant information in this figure is very small, a large part of the figure has no content. In addition, the values for CTR and 0.025 are not clearly visible. An adjustment to improve comprehensibility is recommended.

In addition, the values that do not change at all or only change later (all below 0.10 mg/mL) should also be mentioned in the text.

You can see a clear representation of the morphological changes in Figure 2a but the scaling is missing, also in Figure 7A.

Figure 4 is very small and therefore difficult to read. The size should be oriented to the size of the other illustrations.

line 207-208: Which cell death marker was used? Please add to materials and methods.

Figure 7 is not explained separately (A and B) in the text. Please adapt.

Discussion

In the discussion, the results are sufficiently addressed and discussed on the basis of current literature. Theories are put forward and an outlook for further application of the model is given.

For better understanding of pro-inflammatory processes demonstrated, the results should be summarized in a figure in this section.

Materials and Methods

In general, it should be checked in these sections whether all steps of the experiments carried out and all manufacturers have been mentioned in full. In some places, information seems to be missing.

For example:

    Line 302 to 304

    Line 321: 96-well plates

    Line 339: Primer

    Line 341-347

Line 331-332:

The sentence is incomprehensible. What was used to dilute it?

Line 374-381: The criteria used to classify the score should be presented in a table or similar to improve clarity.

Comments on the Quality of English Language

English language and grammar should be improved.

Reviewer 2 Report

Comments and Suggestions for Authors

Introduction:

1.     The introduction could provide more specific references to support the statements made about Cisplatin's mechanism of action, toxicity, and the release of pro-inflammatory cytokines.

2.     The introduction could briefly mention the potential limitations or challenges of using zebrafish larvae as a model for studying the systemic effects of Cisplatin.

Results:

1.     The result section should follow a logical order, based on the research questions or hypotheses. You should use subheadings or transitions to indicate the different sections of your result section. For example, you can start your result section with "The effect of Cisplatin on larval survival" and then use "The effect of Cisplatin on larval morphology and behavior", "The effect of Cisplatin on cytokine expression", "The effect of Cisplatin on neutrophil distribution", and "The effect of Cisplatin on cell death" to separate the other sections.

2.     The result section should avoid any interpretation or discussion of the data, as this should be done in the discussion section. Instead, the result section should report the data as objectively and accurately as possible. For example, you should not say "This data demonstrated that Cisplatin has dose-dependent effects on mortality when administered systemically on zebrafish larvae." Instead, you should say "Cisplatin induced dose-dependent mortality in zebrafish larvae when administered systemically."

Discussion

1.     The discussion section should be clear and concise, without unnecessary words. For example, you can remove the word "here" from your discussion section.

2.     The discussion section should acknowledge the limitations of the study and suggest directions for future research. It should also state the main conclusion and the take-home message of the study. For example, you can say "Our study has some limitations that should be considered when interpreting the results. First, we used a single dose and time point of Cisplatin exposure, which may not reflect the cumulative and chronic effects of the drug in clinical settings. Second, we did not compare our dilution model with other models or drugs that induce AKI or systemic inflammation, which may limit the generalizability of our findings. Third, we did not measure the levels of Cisplatin or its metabolites in the larvae, which may affect the interpretation of the dose-response relationship and the toxicity mechanisms. Future studies should address these limitations by using different doses and time points of Cisplatin exposure, comparing different models and drugs, and measuring the pharmacokinetics and pharmacodynamics of Cisplatin in zebrafish larvae. Future studies should also explore the molecular and cellular pathways that mediate the effects of Cisplatin on cytokine expression, neutrophil distribution, cell death, and kidney damage, and identify potential targets for intervention. In conclusion, our study shows that Cisplatin induces systemic inflammation in zebrafish larvae, involving increased expression of pro-inflammatory cytokines and migration of neutrophils from the pronephros to other tissues and organs. Our study also shows that Cisplatin induces morphological and behavioral changes and cell death in zebrafish larvae, indicating a broader toxicity spectrum than previously thought. Our study provides a simple and rapid model for studying the role of neutrophils in Cisplatin-induced systemic inflammation, and suggests new avenues for developing therapies or interventions that can modulate the inflammatory response and protect the kidney and other organs from Cisplatin-induced damage. The take-home message of our study is that Cisplatin has different effects on zebrafish larvae depending on the route of administration, and that zebrafish larvae are more sensitive to the drug than adult zebrafish or mammals."

Martials and methods

1.     The materials and methods section should avoid any interpretation or discussion of the data, as this should be done in the results and discussion sections. Instead, the materials and methods section should report the data as objectively and accurately as possible. For example, you should not say "This data demonstrated that Cisplatin has dose-dependent effects on mortality when administered systemically on zebrafish larvae." Instead, you should say "Cisplatin was administered systemically on zebrafish larvae at different concentrations and the mortality was monitored daily."

Comments on the Quality of English Language

The paper could be more concise. Some sentences can be rephrased to remove unnecessary words or phrases

Reviewer 3 Report

Comments and Suggestions for Authors

The presented work describes the study of the influence of cisplatin on the development of inflammation in Danio larvae. Cisplatin was administered directly into the water in which the embryos lived. The impact has been studied  drug on the survival and motility of embryos, on the synthesis of pro-inflammatory cytokines, and on the frequency of neutrophils in their body. The article is an interesting addition to the knowledge about cisplatin toxicity in animal models.

Here are some issues that need to be addressed before an article is accepted:

1.     Headings should be removed from the article abstract in accordance to author’s guidelines.

2.     Cisplatin (cis-diamminedichloroplatinum (II)) should be replaced with proper chemical name cis-diamminedichloroplatinum(II).

3.     Why is cisplatin capitalized throughout this article?

4.     LD50 should be replaced with LD50 (line 96).

5.     In Figure 3, the names should be written in capital letters (IL-12, IL-8, IL-6, IL-10, KIM-1). What do the two triangles and CT mean in the description of the Y axis?

6.     Line 362: should be H2O2.

7.      

Round 2

Reviewer 1 Report

Comments and Suggestions for Authors

The comments have been implemented very well. However, there are still three minor points that need to be improved:

1.  Please check again whether all abbreviations have been explained or add a list of abbreviations, for example:

             Line 55: TNF-α

             Line 79: NF-κB and IRF3

2. Line 126: It is strange that the p-value should be above 0.9999. p>0.05?

3.       Figure 7A: Please indicate the size of the scale bar like in the figures before.
